# A Review of Solid-State Proton–Polymer Batteries: Materials and Characterizations

**DOI:** 10.3390/polym15194032

**Published:** 2023-10-09

**Authors:** M. S. A. Rani, M. N. F. Norrrahim, V. F. Knight, N. M. Nurazzi, K. Abdan, S. H. Lee

**Affiliations:** 1Department of Physics, Faculty of Science, Universiti Putra Malaysia, Serdang 43400, Malaysia; 2Institute of Tropical and Forest Products (INTROP), Universiti Putra Malaysia, Serdang 43400, Malaysia; khalina@upm.edu.my; 3Research Centre for Chemical Defence, Universiti Pertahanan Nasional Malaysia, Kem Perdana Sungai Besi, Kuala Lumpur 57000, Malaysia; victor.feizal@upnm.edu.my; 4Bioresource Technology Division, School of Industrial Technology, Universiti Sains Malaysia, Pulau Pinang 11800, Malaysia; mohd.nurazzi@usm.my; 5Department of Wood Industry, Faculty of Applied Sciences, Universiti Teknologi MARA (UiTM) Cawangan Pahang, Bandar Tun Razak 26400, Malaysia; leesenghua@uitm.edu.my

**Keywords:** proton-conducting, polymer electrolyte, solid-state batteries, characterizations

## Abstract

The ever-increasing global population necessitates a secure and ample energy supply, the majority of which is derived from fossil fuels. However, due to the immense energy demand, the exponential depletion of these non-renewable energy sources is both unavoidable and inevitable in the approaching century. Therefore, exploring the use of polymer electrolytes as alternatives in proton-conducting batteries opens an intriguing research field, as demonstrated by the growing number of publications on the subject. Significant progress has been made in the production of new and more complex polymer-electrolyte materials. Specific characterizations are necessary to optimize these novel materials. This paper provides a detailed overview of these characterizations, as well as recent advancements in characterization methods for proton-conducting polymer electrolytes in solid-state batteries. Each characterization is evaluated based on its objectives, experimental design, a summary of significant results, and a few noteworthy case studies. Finally, we discuss future characterizations and advances.

## 1. Introduction

Recent years have witnessed a significant focus from both academic and industrial researchers on the interdisciplinary field of polymer electrolytes. This field spans various disciplines, including polymer physics, electrochemistry, organic chemistry, and inorganic chemistry, all aimed at advancing electrochemical devices like batteries, supercapacitors, fuel cells, solar cells, sensors, and more [1,2,3,4]. Traditional liquid electrolytes have been prevalent in electrochemical power sources due to their high ionic conductivity. However, they come with inherent limitations, including low electrochemical stability, corrosive interactions with electrodes, and leakage, making them unsuitable for use in many electrochemical devices [5,6]. Additionally, safety concerns arise from dendrite growth in aqueous electrolytes during charge/discharge cycles in rechargeable cells, which can lead to internal short circuits in batteries. Consequently, extensive research on solid polymer electrolytes is underway to identify suitable materials that can replace liquid-based electrolytes. Solid polymer electrolytes offer several advantages, such as improved electrode–electrolyte interface contact, ease of processing, low self-discharge in batteries, excellent elasticity, absence of leakage, high safety levels, and superior mechanical and adhesive properties [7,8,9]. To date, a wide range of polymer materials has been explored as hosts for creating these electrolytes.

The development and improvement of energy storage systems are increasingly crucial in response to the growing demand for sustainable energy sources [10]. Currently, lithium-ion (Li^+^) batteries stand out as one of the most promising technologies, especially for vehicular applications. This is primarily due to the small ionic radius of Li^+^ ions, enabling their intercalation into the layered materials within electrodes [11]. However, concerns have arisen regarding the supply of lithium, as it is both costly and primarily concentrated in geographically distant or politically unstable regions [12,13]. Even with the implementation of significant battery recycling programs, it may not be sufficient to prevent the depletion of lithium resources promptly [14]. Moreover, the increased demand for lithium in medium-scale automobile batteries could drive up the price of lithium compounds, making large-scale energy storage economically unfeasible [15,16]. As an alternative, researchers are actively seeking low-cost and readily available elements for practical applications. Among the candidates for providing H^+^ ions, ammonium salts have emerged as particularly promising choices and have garnered significant interest as ionic dopants in polymer electrolytes.

One of the primary challenges in applying solid-state polymer science to proton batteries is enhancing the proton conductivity in polymer electrolytes. Researchers are actively exploring various strategies to improve the ionic conductivity of polymer matrices. These strategies include incorporating proton-conducting additives, such as acids or hydrated metal ions, into the polymer structure [17,18]. Additionally, they are designing and synthesizing novel polymer architectures with optimized proton transport properties. Furthermore, researchers are working on enhancing the mechanical properties of solid-state polymer electrolytes to ensure stability and flexibility. The mechanical strength of the electrolyte can be improved by incorporating reinforcing fillers or designing cross-linked polymer networks [19,20,21]. This is crucial for practical applications, as solid-state electrolytes must maintain their integrity and performance under varying operating conditions. Another area of focus in the field of solid-state polymer science for proton batteries is the development of polymer-electrode materials. Conventional lithium-ion battery electrodes typically rely on inorganic materials like graphite or transition-metal oxides. However, these materials may not meet the requirements for proton batteries due to differences in charge-carrier needs. Researchers are exploring polymer-based electrode materials, such as conductive polymers or polymer composites, as alternatives to enable efficient proton storage and transport.

To the best of our knowledge, no specific reviews on proton-conducting polymer electrolytes in solid-state batteries have been reported to date. It is intriguing to dedicate a review primarily to the properties of proton-conducting polymer electrolytes, particularly those prepared using simple and cost-effective methods like the conventional solution-cast technique. Although the field of proton batteries is relatively small at present, there is a significant opportunity to pave the way for new applications, especially in the realm of small portable electronic devices operating within the 1.5 V range. It is of paramount importance to comprehensively review the material characterization and performance aspects of proton batteries. This compilation of articles aims to create a substantial database for researchers investigating the properties of proton-conducting polymer electrolytes and their potential applications.

This review aims to provide a timely overview of the development of proton-conducting-polymer electrolyte batteries reported in the literature surveyed from 2011 to 2023. We focus on the recent progress in the characterization technique of choice for proton-conducting polymer-electrolyte (PCPE) batteries. However, a few older papers are also cited to convey a basic understanding and to draw comparisons. The review of proton-conducting polymer electrolytes for solid-state batteries is organized into three parts:An introduction to proton batteries: components, chemistry, and materials.The electrochemical characterization of materials: morphologies, structural, ionic conductivity, electrochemical as well as thermal properties, and,Performance of PCPE in solid-state batteries: performance characterization, such as open-circuit voltage (OCV), current–voltage (*I-V*), power density–current density (*J-P*), discharge, and charge–discharge profile.

Each characterization technique is followed by a review of related examples. The properties of PCPE will be reviewed in detail and future research directions are also discussed. 

## 2. Components and Mechanism of Proton Battery-Based Polymer Electrolytes

The standard configuration and operation principle for battery-based PCPE involves three main components (Figure 1d):i.Anode (Figure 1a): Oxidation occurs (electrons flow to the external circuit). The anode must be an efficient oxidizing agent, stable in adhesion with electrolyte, and have a useful working voltage, long lifetime, high reversible discharge capacity, and low surface area for safety improvement [22], i.e., combinations of zinc (metal powder), ZnSO_4_⋅7H_2_O and graphite powder [23,24].ii.Cathode (Figure 1b): Reduction occurs (positive terminal of the battery in the discharged mode). i.e., mixtures of lead oxide (PbO_2_) [25,26], vanadium pentoxide (V_2_O_5_) (active cathodic material) [27,28], graphite (provides an adequate electronic conductivity), manganese dioxide (MnO_2_) [29,30] and small ratio of polymer electrolyte used in the system (to favor electrode/electrolyte interfacial contact and helps in reducing electrode polarization) [31].iii.Electrolytes (Figure 1c): The medium (ions are transferred between the anode and cathode during charge and discharge). Electrolytes with high ionic-conductance (range between 10^−5^ to 10^−2^ S cm^−1^), high thermal and chemical stability, wide potential window (defined as the range in voltage between the oxidative and reductive decomposition limits of the electrolyte), low reactivity toward other components in the battery and have ionic transference number greater than 0.9 are suitable for battery applications [32], i.e., the use of various type of polymer electrolytes (solid; chitosan [33,34], poly (ethylene oxide) [35,36], liquid; (1,1,2,2-tetrafuoroethyl-2,2,3,3-tetrafuoropropyl ether (TTE) [37] and gel; poly(vinylidene fluoride-co-hexafluoropropylene)-ionic liquid [38]). Table 1 summarizes the ionic conductivities at an ambient temperature of some PCPE systems.

A PCPE consists of positively charged protonic species that include H^+^, H_3_O^+^, and NH_4_^+^. Proton conductors operate like other types of polymer electrolytes. In an electrochemical device, they feature separate anode and cathode components, impede electronic conduction, and facilitate the transfer of charges in the form of desired ions. This electrochemical cell is typically enclosed between two stainless-steel sheets, which serve as current collectors. The battery is configured as follows,
Zn + ZnSO_4_·7H_2_O + AC (anode)||PCPE (electrolyte)||MnO_2_ + AC (cathode)(1)

The chemical reaction that possibly occurred in the proton battery is [39],

i.At the anode, Zn was oxidized with the release of two electrons and ZnSO_4_·7H_2_O is the donor of H^+^ ions.
Zn → Zn^2+^ + 2e^−^ E_ox_ = 0.76 V(2)
ZnSO_4_·7H_2_O → 7H^+^ + 7OH^−^ + ZnSO_4_ E_ox_ = −0.82 V(3)
ii.At the cathode, MnO_2_ was reduced with the acceptance of electrons.
MnO_2_ + 2e^−^ + 4H^+^ → Mn^2+^ + 2H_2_O E_red_ = 1.22 V(4)
iii.The overall reaction in the cell is
E_ox_ + E_red_ = E_cell_(5)
Zn + ZnSO_4_·7H_2_O + MnO_2_ + 2e^−^ + 4H^+^ → Zn^2+^ + 7H^+^ + 7OH^−^ + ZnSO_4_ + Mn^2+^ + 2H_2_O
−(0.76 − 0.82) V + 1.22 V = 1.28 V


Proton batteries involve a reaction where ions are inserted into a cathode material, often structured in a layered lattice. This process is known as intercalation, where ions move between the layers. During intercalation, the cathode materials expand, causing the surrounding solid material to flex and maintain contact, even as the particles change in size and shape. Typically, the cathode material exhibits poor electronic conductivity. To facilitate the flow of electrons to and from the current collector, carbon is often added to the cathode particles. 

**Figure 1 polymers-15-04032-f001:**
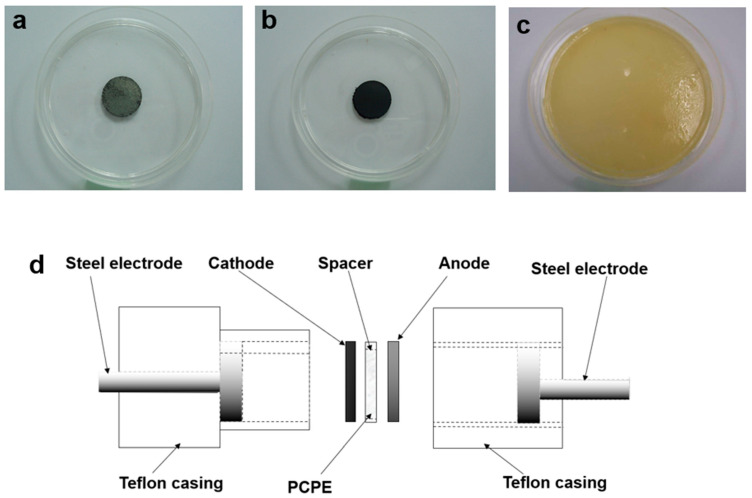
Photograph of (**a**) anode (**b**) cathode (**c**) PCPE film and (**d**) schematic diagram of PCPE batteries.

**Table 1 polymers-15-04032-t001:** Examples of polymer-salt complexes in PCPEs system and their conductivity.

Electrolyte System	State	Electrochemical Properties				Ref.
		Ionic Conductivity σ (S cm^−1^)	I-TN	Stability (V)	E_a_	
Tapioca starch/PEO-NH_4_NO_3_	solid	2.8 × 10^−7^	-	-	-	[40]
Sago starch-NH_4_Br		6.9 × 10^−9^	-	-	0.07 eV	[41]
Potato starch-NH_4_I		2.4 × 10^−4^	0.95	-	-	[42]
Corn starch/Chitosan NH_4_I-glycerol		1.3 × 10^−3^	0.99	1.9	0.18 eV	[43]
Starch/Chitosan NH_4_I		3.0 × 10^−4^	-	-	0.20 eV	[44]
Starch/Chitosan NH_4_Cl-glycerol		5.1 × 10^−4^	-	-	0.19 eV	[45]
Starch/Chitosan NH_4_Br-EC		1.4 × 10^−3^	0.92	0.18	0.17 eV	[46]
Rice starch NH_4_I		1.4 × 10^−4^	-	-	-	[47]
Chitosan acetate—NH_4_CF_3_SO_3_—DMC		~10^−6^	-	-	0.60 eV	[48]
Chitosan acetate/PEO—NH_4_NO_3_		1.0 × 10^−4^	-	-	-	[49]
Chitosan/PEO—NH_4_I–I_2_		4.3 × 10^−6^	-	-	-	[50]
Cellulose acetate—NH_4_SCN		3.3 × 10^−3^	0.99	-	0.15 eV	[51]
Chitosan/PEO—NH_4_NO_3_—EC		2.1 × 10^−3^	-	1.75	0.18 eV	[52]
MG-30–NH_4_CF_3_SO_3_—EC		~10^−4^	-	-	-	[53]
PEO-NH_4_ClO_4_		~10^−5^	-	-		[54]
carboxymethyl *kappa*-carrageenan/CMC-NH_4_I		2.41 × 10^−3^	-	-	-	[55]
Alginate-(NH_4_)_2_SO_4_		3.01 × 10^−5^	-			[56]
Agar-NH4_S_CN		1.0 × 10^−3^	-	-	-	[57]
Dextran-NH_4_Br		1.67 ± 0.36 × 10^−6^	0.92	1.62	-	[58]
Chitosan/PEO–NH_4_I–I_2_–[BmIm][I]	gel	5.5 × 10^−4^	-	2.5	0.17 eV	[59]
Gelatin-HCL-glycerol		5.4 × 10^−5^	-	-	-	[60]
Gelatin–Acetic acid–glycerol		8.7 × 10^−4^	-	-	-	[60]
MG-49–NH_4_CF_3_SO_3_–PC		2.9 × 10^−2^	-	-	-	[61]
Gellan–H_2_SO_4_		1.5 × 10^−3^	-	-	0.17 meV	[62]

## 3. Proton-Conducting Polymer-Electrolyte Characterizations and Properties

Characterizing PCPEs is paramount in the quest to identify the ideal composition for application in solid-state batteries. These characterizations aim to pinpoint the most favorable properties, which encompass achieving the highest ionic conductivity, optimizing surface morphology for excellent electrode/electrolyte interfaces, fine-tuning the polymer host structure, and ensuring the utmost thermal stability in PCPE films. 

### 3.1. Ionic Conductivity and Temperature Dependence Analysis

Ionic conductivity stands as the paramount factor in advancing PCPE research, often guiding the selection of the most conductive PCPE for solid-state battery fabrication. The characterization of a PCPE film involves placing the sample between two symmetrical blocking electrodes, such as stainless steel 316 [18,63], platinum [64], brass [65], or gold [66] (Figure 2a). In the case of PCPE gel or liquid electrolyte, the sample is deposited within the spacer ring between the blocking electrodes inside a Teflon cell (Figure 2b) [67]. Once the sample is connected to the electrochemical impedance spectroscopy (EIS) instrument, it undergoes testing with modest amplitude signals [68,69,70] at specified frequency ranges [70,71,72,73,74,75,76,77]. Subsequently, the data are analyzed using specialized software. It is calculated using the bulk resistance (*R*_b_) value from the Nyquist plot. The ionic conductivity of a PCPE is presented in four ways which are Nyquist plot (the semicircle/spike line from this plot is used to calculate the value of *R*_b_) (Figure 2c), equivalent circuit model (presents the EIS circuit model) (Figure 2d), ionic conductivity graph (the value and behavior of the polymer host) (Figure 2e) and temperature dependence graph (to determine the ionic behavior and activation energy) (Figure 2f). There are many ways to enhance ionic conductivity such as blending, doped with ionic salts, incorporation with plasticizers and ionic liquids, etc. [29]. Normally, the optimized ionic conductivity of PCPE with magnitude orders of 10^−5^ to 10^−2^ S cm^−1^ is chosen as electrolytes in the fabrication of solid-state batteries [25,30,31,78,79,80].

### 3.2. Morphology Properties

Scanning electron microscopy (SEM) with magnification ranging from 10 to 100 k [85] or field emission scanning electron microscopy (FESEM) with magnification between 10 and 300 k [86] (Figure 3a) [87,88,89] is commonly employed in PCPE research. These techniques allow for cross-section analysis, offering insights into the sample’s interior, shape, size, and grain size, as well as surface roughness in PCPE cross-sections. SEM and FESEM serve as additional confirmations for X-ray diffraction pattern results. For example, an investigation into chitosan starch-0–50 wt.% sodium iodide (NH_4_I) revealed a porous structure when 40 wt.% NH_4_I was added [44]. This porosity enhances ionic conductivity, as pore connectivity is vital for charge-carrier transportation in PCPE. However, the presence of solid particles suspended above the surface, caused by salt recrystallization, can lead to the loss of charge carriers [13,89]. SEM cross-section analysis was performed on carboxylated chitosan hydrogel to confirm the absence of obvious composition separation, indicating good compatibility between carboxylated chitosan hydrogel and hydrochloric acid (Figure 3b) [90]. The obtained cross-section displayed a compact, non-porous structure, which facilitates the loading and maintenance of electrolyte ions through the free volume of flexible polymer chains [91,92]. In summary, the development of efficient proton batteries necessitates samples with smooth surfaces (ensuring excellent contact between the electrode–electrolyte interface), uniform surfaces to facilitate smooth ion transport along the polymer chain, and appropriate pore sizes in the film.

### 3.3. Structural Characterization 

Crystallinity is a state of molecular structure referring to a long-range periodic geometric pattern of atomic spacing. A structural characterization is a detailed description of the PCPE nature (crystalline/semi-crystalline structure). Understanding structural properties is critical to determining the phase composition/changes, and degree of crystallinity (*X*_C_ or crystallinity index) in the crystalline structure of polymers. One of the simplest methods to monitor the pattern in crystallinity of polymers is via X-ray diffraction studies. The PCPE is irradiated with a monochromatic Cu/Kα1 irradiation beam (single wavelength = 1.5406 Å) at 2θ angles between 10° to 60° to produce a regular pattern of reflection. The intensity of the X-rays dispersed from the entire sample is the total area under the diffraction pattern, which is divided into two regions (crystalline and amorphous) [96]. The deconvolution method is also employed to obtain the *X*_C_ [97]. Previous studies have shown that various types of PCPE with high conductivity (magnitude of 10^−5^ to 10^−3^ S cm^−1^) and an amorphous structure are preferable materials for rechargeable battery applications [95,98,99]. The strategies described modifications that can be implemented to change the structure/reduce the crystallinity of the PCPE are blending of polymers [94,100] (Figure 3c), addition of cross-linker [101,102,103], incorporation of ionic salts [34,104,105], addition of fillers/additives [101,106,107] and addition of plasticizers [81,108,109] (Figure 3d).

### 3.4. Thermal Properties 

Thermal characterization refers to a set of analytical techniques for determining how the PCPE properties change as a function of temperature. TGA (Figure 3e) was applied to determine the thermal stability and degradation of PCPE while DSC (Figure 3f) is a vital analysis to determine the glass transition temperatures (*T*_g_) of the sample [110]. To conduct a TGA analysis, a small amount of the sample was placed in an aluminum/platinum pan holder and was heated in a nitrogen atmosphere for a certain rate (i.e., 10 °C min^−1^) [76,111] from room temperature to various temperatures (600, 800, 1000 °C) [69,110,112,113,114] while to conduct DSC analysis, the sample is heated at (i.e., 10 °C min^−1^) [115] in a nitrogen atmosphere and operates in a low-temperature range (<500 °C, typically ranging from room temperature to 250 °C) [68,115]. The PCPE amorphicity increases with the addition of ionic salt/plasticizers/ionic liquids (as confirmed by structural analysis) reducing the decomposition temperature compared to crystalline materials [116]. To produce a good electrolyte for solid-state batteries, an electrolyte should have a low *T*_g_ (reduction of transient cross-linkage between the oxygen atom and proton which leads to the softening of the polymer) [25,80,117].

## 4. Linear Sweep Voltammetry

The anodic or oxidative stability of an electrolyte plays a crucial role in defining the effective ‘voltage window’ of a battery. This voltage window represents the highest voltage to which a cell can be cycled without causing electrolyte decomposition. It serves as a critical metric for calculating the maximum cathode voltage that can be safely cycled without adverse effects or electrolyte deterioration. Determining the decomposition voltage of the electrolyte can be achieved through a technique known as linear sweep voltammetry (LSV). In LSV, a series of potential sweeps occur linearly with time, ranging from a lower limit to an upper limit. The higher the breakdown voltage observed in LSV, the better the electrochemical stability of the electrolyte. In this section, we will summarize various factors that influence the electrochemical stability of PCPE in the context of solid-state proton–polymer battery fabrication. These factors include the preparation method and the addition of specific materials.

In terms of setup, the LSV characterization of PCPE films follows a similar configuration to the electrochemical impedance spectroscopy (EIS), with SS316 serving as the anode, PCPE as the electrolyte, and SS316 as the cathode. However, it is important to note that the parameters used in LSV may differ from those in EIS (Figure 4a). In the case of liquid or gel PCPE systems, a 3-electrode cell setup can be employed. In this setup, electrode rods are immersed in liquid or gel. Typically, inert materials like stainless steel 316, platinum, and others are selected as electrodes [118,119]. To measure the breakdown voltage within a specific range, the amplitude signal input is replaced by a scan rate input, such as 10 mV s^−1^ [120], 5 mV s^−1^ [78], and 1 mV s^−1^ [77]. Once the cell is assembled, a typical LSV plot for an asymmetric cell is scanned up to a high positive voltage, for example, 5.0 V (Figure 4b). During the voltage sweep in the high-voltage region, the onset of a decomposition phenomenon is indicated by a sharp increase in current. This onset typically occurs at 2.5 V, followed by a gradual increase, and then an exponential rise after 4.0 V. A more precise determination of the onset can be obtained by identifying the x-intercept of the tangent to the first step of the current increase. To establish the electrochemical window, an arbitrary current density threshold is set, often at 0.1 mA cm^−2^ [121]. However, it is important to note that the onset of the current can be influenced by several factors, including reaction volume, voltage scan rate, temperature, and the material of the current collector.

LSV studies on PCPE films have reported outcomes influenced by different concentrations of ionic salts and plasticizers. For instance, a recent study involving a chitosan/PVA blend film with 40 wt.% NH_4_I showed an interception at 1.33 V in the LSV response, indicating electrolyte decomposition (Figure 4c) [87]. This potential cut-off from the LSV response suggests that the electrolyte is suitable for use in proton-based energy devices. In another report, the LSV voltammogram of carboxymethyl cellulose doped with NH_4_SCNrevealed electrolyte degradation occurring at 1.6 V (Figure 4d) [120]. This outcome confirms that the film experiences electrolyte degradation at the surface of the stainless-steel electrode when the potential exceeds the limit [122]. Such reactions demonstrate film instability, leading to capacity fading and irreversible reactions. Consequently, for use in protonic energy devices, meeting the requirement of a minimum decomposition potential of 1 V is crucial [78,84]. Additionally, the LSV voltammogram of the highest conducting chitosan:potato starch blend with NH_4_F electrolyte, recorded at 100 mV/s, shows that as the potential reaches 1.78 V, the current begins to rise dramatically. This onset value is also satisfactory for applications in proton-based devices [123]. Another investigation into electrolyte decomposition involved chitosan:dextran-NH_4_Br film, conducted by Aziz and colleagues using stainless steel as working electrodes, a scan rate of 10 mV/s, and ambient temperature conditions [33]. The film demonstrated decomposition at a voltage greater than 1.54 V, highlighting its stability across a wide range of potential windows [31,95,98].

A few researchers discovered that adding a plasticizer to a PCPE system can improve their electrochemical properties [29,39,92,124,125]. The breakdown voltage value is not significantly different from that of a chitosan–polymer blend system. The LSV voltammogram for chitosan: methylcellulose-NH_4_SCN-42 wt.% glycerol (Figure 4e) [126]. The current increased when the potential reached 2.11 V, indicating that the ionic conductivity influenced the decomposition voltage of the chitosan: methylcellulose-NH_4_SCN–glycerol electrolyte. Another interesting finding was found where the breakdown voltage for the glycerolized chitosan NH_4_F zinc metal is higher compared to glycerolized chitosan NH_4_F at 10 mV/s. This demonstrates that the addition of the zinc complex can broaden the potential window of the glycerolized electrolyte. Despite concerted efforts, the best oxidative stability for a proton battery electrolyte that could be achieved experimentally is greater than 3 V, while the minimum electrochemical window standard for a protonic battery is about ~1 V.

The following conclusions can be drawn from the findings on the breakdown voltage of PCPE via LSV, 

i.LSV: a voltammetric technique for determining breakdown voltage/decomposition/electrochemical stability of PCPE.ii.Factors affecting onset current: reaction volume, voltage scan rate, temperature, and the material of the current collector.iii.The minimum decomposition potential of a PCPE is 1 V (implying that the PCPE is appropriate for use in solid-state polymer batteries).

**Figure 4 polymers-15-04032-f004:**
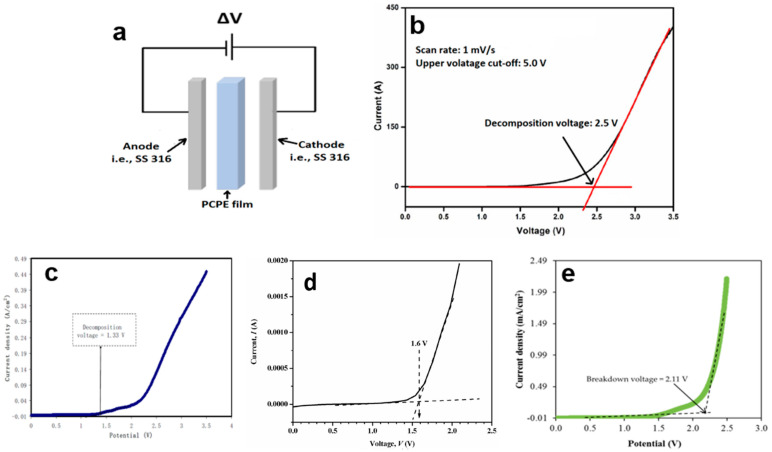
(**a**) Setup of LSV characterization, (**b**) sample of LSV voltammogram, LSV voltammogram of (**c**) poly(vinyl alcohol): chitosan NH_4_I [87], (**d**) CMC-NH_4_SCN. Adapted from ref. [120], (**e**) chitosan: methyl cellulose-NH_4_SCN glycerol [126].

## 5. Cyclic Voltammetry

Cyclic voltammetry (CV) is a highly versatile electroanalytical technique extensively employed in various fields, including electrochemistry, inorganic chemistry, organic chemistry, and biochemistry, owing to its versatility, ease of measurement, and effectiveness. CV provides a valuable tool for rapidly observing redox behavior over a wide potential range [127]. In essence, CV can be viewed as an extended version of linear sweep voltammetry (LSV), as it involves scanning the voltage in both directions after sweeping across a predefined range. During these voltage sweeps, the current response is recorded. Comparing the shape and magnitudes of the currents provides valuable insights into the reversibility of the reaction. Although good ionic conductivity is a critical factor for PCPE, it alone may not suffice to establish its suitability for the fabrication of electrochemical devices. This underscores the significance of characterizing PCPE through CV. Such characterization is essential because the stability of a broad electrochemical window is a critical requirement for the practical application of solid-state polymer batteries [128].

To confirm the protonic conduction in the PCPE, CV has been applied. This characterization normally has been carried out on the following two symmetric cells, i.e., Cell 1 (Figure 5a): SS316 (ion blocking electrode)|PCPE|SS316 (ion blocking electrode) and Cell 2 (Figure 5b): Zn + ZnSO_4_·7H_2_O (reversible electrode)|PCPE|Zn + ZnSO_4_·7H_2_O (reversible electrode) [78,118,129,130]. The reversible electrode acts as an H^+^ source by the given equation:Zn + ZnSO_4_∙7H_2_O ↔ Zn^2+^ + 2e^−^ + ZnSO_4_ + 7H^+^ + 7OH^−^(6)

To measure protonic conduction within a specific range, the amplitude signal was substituted with a scan rate input, commonly set at values such as 50 mV s^−1^, 10 mV s^−1^, and 5 mV s^−1^ [128,131,132,133]. It is worth noting that this characterization step is typically conducted before evaluating the long-term cyclability of PCPE when considering its suitability as a component in a full-cell battery setup.

CV studies on PCPE have reported voltammograms for different concentrations of ionic salt and plasticizers. For instance, comparative CV voltammograms were obtained for Cell 1 (SS316 electrodes) and Cell 2 (Zn + ZnSO_4_·7H_2_O electrodes) using the highest conducting PVA + 80 wt.% [BMIM][HSO_4_] electrolyte at a scan rate of 10 mV/s (Figure 5c) [134]. In the Cell 2 voltammograms, two distinct cathodic and anodic current peaks are evident, whereas Cell 1 shows no such feature. This observation suggests the occurrence of cathodic deposition and anodic oxidation at the electrode/electrolyte interface in Cell 2. The anodic and cathodic peaks in the voltammogram of Cell 2 are separated by approximately 1.3 V due to the two-electrode geometry with no reference electrode. Additionally, the magnitude of currents in Cell 2 is greater than that in Cell 1, further supporting the presence of proton conduction in the system.

Some researchers have found that incorporating a plasticizer into PCPE can enhance protonic conduction in the system [128,133,134]. To confirm protonic conduction in a plasticized system (PEO-NH_4_PF_6_-polysorbate 80), CV techniques have been employed [132]. At a scan rate of 10 mV/s, comparative CV plots were generated for Cell 1 (SS electrodes) and Cell 2 (Zn electrodes). In the CV plots for Cell 2, two distinct cathodic and anodic current peaks are observed, separated by approximately 2.5 V. It is important to note that the shifting of the positions of these peaks is a result of the two-electrode geometry with no reference electrode. In contrast, no such features are observed in Cell 1. The voltammogram of Cell 2 demonstrates the cathodic deposition and anodic oxidation of protons at the electrode–electrolyte interface, providing strong evidence for protonic conduction in PCPE.

A recent study investigated the impact of SiO_2_ nano-filler on the protonic conduction of PCPE comprising polyvinyl alcohol (PVA) and 1-butyl-3-methylimidazolium hydrogen sulfate [C_4_C_1_Im][HSO_4_] ionic liquid [131]. The findings align with our previous discussions. CV plots reveal protonic oxidation and reduction at the respective electrode–electrolyte interfaces of Cell 2 Zn + ZnSO_4_∙7H_2_O|PVA-[C_4_C_1_Im] [HSO_4_]-SiO_2_]|Zn + ZnSO_4_∙7H_2_O), confirming that the addition of SiO_2_ nano-filler enhances protonic conduction in PCPE. In another study, the CV of a porous chitosan membrane was conducted. Interestingly, researchers discovered that the electrochemical stability of the dry porous chitosan membrane (3.3 V) slightly increased to 3.8 V after being soaked in CH_3_COONH_4_ for 48 h (Figure 5d). This value surpasses the breakdown voltage (1.75 V) since the calculation of electrochemical stability was based on the width of the electrochemical window. The enhanced electrochemical stability can be attributed to the high absorbency of H^+^ ions provided by CH_3_COONH_4_ within the porous chitosan membrane. However, beyond the electrochemical window, the evolution of oxygen and hydrogen gases commences, leading to membrane degradation.

The following conclusions can be drawn from the findings of PCPE via CV: i.CV is a valuable technique for confirming protonic conduction in PCPE, complementing complex impedance spectroscopy.ii.Protonic conduction in PCPE is enhanced through various means, including the addition of ionic salts, plasticizers, ionic liquids, fillers, and other factors.

**Figure 5 polymers-15-04032-f005:**
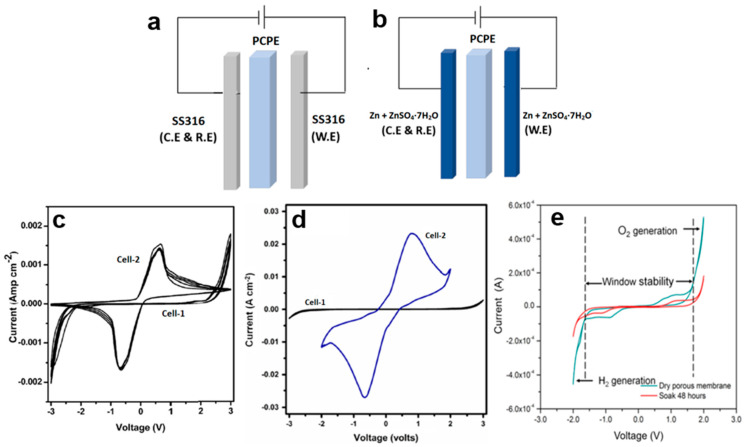
Configurations of (**a**) Cell 1 (**b**) Cell 2, and cyclic voltammetry of (**c**) Cell 1 (SS|PVA + 80 wt.% [BMIM][HSO_4_]|SS), Cell 2 (Zn + ZnSO_4_∙7H_2_O|PVA + 80 wt.% [BMIM][HSO_4_]|Zn + ZnSO_4_·7H_2_O) [134], (**d**) Cell 1 (SS|PVA-[C_4_C_1_Im][HSO_4_]-SiO_2_]|SS), Cell 2 (Zn + ZnSO_4_∙7H_2_O|PVA-[C_4_C_1_Im][HSO_4_]-SiO_2_]|Zn + ZnSO_4_∙7H_2_O), (**e**) dry porous membrane before and after being soaked in ammonium acetate for 48 h, where the window stability was from −1.5 V to 1.8 V.

## 6. Open-Circuit Voltage

Open-circuit voltage (OCV) is also known as open-circuit potential (OCP) or voltage potential (*V*_oc_). OCV is the potential difference between two terminals of a battery that are not linked to any circuit, implying that no external electric current flows through the battery. The OCV value reflects the full potential of a battery since the potential does not share any of its potential with a load (at rest). This technique is beneficial for determining the potential range of proton battery fabrications.

The OCV is typically measured with a True RMS Multimeter (Figure 6a). To run the OCV test, the red crocodile clip is connected to the anode—the black one to the cathode, and the white one to the reference electrode. Then, the OCV technique is chosen in the program, and it is implemented by selecting the desired parameters of time, higher and lower voltage, and scanning rate. The proton battery was stored and monitored in open-circuit mode for a few hours. The proton battery could be stabilized in a matter of hours so that the voltage remained consistent.

OCV plots are typically categorized into two types: (a) voltage versus time and (b) voltage versus current density plots. High-quality proton batteries maintain a consistent voltage over extended storage periods. Initially, the OCV plot may experience a slight decrease due to the conditioning effect of the proton batteries, but it remains stable for an extended duration of storage (Figure 6b). Interestingly, the OCV value is determined at the start of the voltage versus current density graph (as seen in an H_2_/air fuel cell) [135] (Figure 6c). The gradual voltage decline observed in this plot is attributed to the polymer electrolyte’s increased absorption of water from its surroundings.

The OCV of proton batteries is influenced by the polymer electrolyte. Previous studies have explored various techniques to improve the OCV of proton batteries. Doping with salts containing high concentrations of H^+^ ions, such as NH_4_NO_3_ and NH_4_CH_3_COO, has been conducted [136]. Additionally, the incorporation of plasticizers like ethylene carbonate and glycerol [83], or fillers like Al_2_O_3_ and SiO_2_ [89], has not only enhanced electrolyte conductivity but also improved the OCV over the long term. Another approach to increase the OCV value (from 1.5 to 1.6 V) is modifying the electrolyte by combining different types of polymers, such as PVA, PMMA, PVDF, cellulose, or starch [125,129]. Changes in the polymer electrolyte, such as using gel polymer electrolytes (GPE), have also impacted the OCV (reaching 1.41 V) and maintained it for 48 hours of storage. Although the OCV value may not exceed 1.6 V, GPE can remain in a liquid state for a reasonable duration, serving as a medium for charge transfer during storage. The application of a porous chitosan–polymer electrolyte in proton batteries has similarly enhanced the OCV and extended the liquid state duration. Differences in OCV values can also be observed when comparing two different cathodes, MnO_2_ and V_2_O_5_, at various temperatures, ranging from 25 to 80 °C. Notably, at the highest temperature (80 °C), the OCV value decreased due to damage to the components in the proton batteries after 33 hours (Figure 6d).

The OCVs of proton batteries based on proton-conducting polymer electrolytes are summarized as follows:i.The OCV characterization: (a) voltage versus time, (b) voltage versus current density at different temperatures,ii.The OCV values of proton batteries based on polymer electrolytes: 1.3–1.7 V,iii.Techniques to improve OCV value and storage duration: modifications to the polymer electrolyte by the addition of salts, plasticizers, fillers, and by blending with another type of polymer.

## 7. Current–Voltage and Power Density–Current Density

The current-voltage (I-V) graph is a plot of voltage as a function of different drain currents. Typically, the specified current drain falls within a range between microamperes (μA) to milliamperes (mA), such as from 5.0 μA to 100.0 mA, or 1.6 μA to 35.0 mA [78]. Higher current values result in a more rapid reduction in voltage. If the current is too high, it can complicate the analysis and potentially damage the components of the proton batteries.

A typical *I-V* graph typically exhibits linearity, indicating that electrode polarization primarily stems from ohmic contributions. The current density-power density (*J-P*) plot is constructed using secondary data derived from the *I-V* plot. This plot helps determine the maximum power density (*P*_max_). High conductivity, along with improved electrode–electrolyte contact, can increase the Pmax value. Another useful parameter obtained from the *J-P* plot is the short-circuit current density (*J_SC_*). To calculate *J_SC_*, the lowest current density (*J*) value in the *J-P* plot is divided by the electrode’s area. However, some reports calculate ISC (short-circuit current) using only the *I-P* plot without considering the electrode area. Several factors influence the Pmax and *J_SC_* values in proton batteries, including the types of salts mixed with the polymer host (e.g., NH_4_Cl and NH_4_NO_3_), the use of plasticizers (e.g., glycerol [129] and EC [26]), and the incorporation of other polymers (e.g., PVA [25]). The choice of cathode material, such as MnO_2_ and V_2_O_5_, can also significantly impact the interfacial resistance between the electrolyte and the electrode.

An important result obtained from the *I-V* plot is the internal resistance (*r*) of the proton battery, which is determined by calculating the slope of the voltage-current curve. In general, a good battery is characterized by a lower *r*, as a higher *r*-value can reduce the maximum power density (*P*_max_) of the battery. For instance, when comparing two proton batteries using different electrolytes, one based on a chitosan NH_4_NO_3_-EC electrolyte exhibited a lower *r*-value compared to another using a starch–chitosan NH_4_Cl-glycerol electrolyte [78] (Figure 7a,b). However, there are criticisms regarding the method and interpretation of *r* values. The *r*-value is typically seen as representing the properties of the anode, cathode, and electrolyte. Nevertheless, the *r* obtained through this calculation encompasses all components, including external circuit resistance. Therefore, it cannot be solely attributed to the anode, cathode, or electrolyte. Factors that can influence the value of *r* include the type and size of the anode, as well as elements that may cause rapid corrosion. High *r* values are often associated with interfacial resistance between the electrolyte and the electrode.

Modifying a chitosan solid polymer electrolyte (SPE) to create a porous membrane also led to a reduction in the *r*-value. The presence of pores within the membrane decreases the interfacial resistance between the electrolyte and the electrode, resulting in a lower *r*-value (Figure 7c). Furthermore, the lower *P*_max_ achieved by this coin-sized proton battery demonstrates that *P*_max_ is influenced by the size of the anode. Although substituting a chitosan GPE was expected to reduce interfacial resistance compared to the SPE, the *r*-value remained higher, and the Pmax value was reduced. Additionally, a study systematically investigated the influence of temperature on proton batteries using separate *I-V* and *J-P* plots (Figure 7d,e). The results indicated that proton batteries operate optimally at a temperature of 60 °C.

The following are the details of the *I-V* and *J-P* graphs of proton batteries:i.The *I-V* plot: small current draws ranging from 5.0 A to 100.0 mA were utilized.ii.The *r*-value of the battery: should be as low as possible since a higher *r* will reduce the Pmax of the battery.iii.*J*_SC_: the lowest value of *J* in the *J-P* plot.iv.Modifications that can be made to reduce the value of r and increase the *P*_max_: (a) blending a few different types of polymers, (b) using GPE, (c) reducing the size of the battery and (d) varying the type of cathode.

## 8. Charge–Discharge Profile

During the discharge process, positive ions migrate from the anode to the cathode through the electrolyte, while negative ions move in the opposite direction. This migration of ions leads to the generation of cell voltage because the anode accumulates a negative charge, and the cathode accumulates a positive charge. Simultaneously, electrons flow through the external circuit from the anode to the cathode, creating an electric current in the opposite direction. The reverse process occurs during charging. Specifically, during the discharge process, an oxidation reaction takes place at the cathode, where the cathode material releases positive ions that flow through the electrolyte. Conversely, a reduction reaction occurs at the anode, where electrons are consumed by transferring positive ions from the electrolyte. These charging and discharging processes are illustrated graphically in Figure 8a.

The produced cell is subjected to various charge and discharge currents using battery test systems. These testing systems typically have dedicated channels where batteries can be connected, and specific testing protocols can be selected within the computer program to initiate the tests. During testing, the battery’s performance is monitored at regular intervals to check for potential issues such as short circuits or deviations in voltage, current, or capacity. The charge–discharge tests are customized to cover a wide range of current levels, typically ranging from 1 mA to 5 A, and voltage levels spanning from 5 to 15 V. Simultaneously, the software displays the results in the form of various plots for specified cycles, including graphs depicting voltage and current vs. time, cell voltage vs. capacity, charge/discharge capacity vs. cycle, and more. Detailed data for each cycle step is also provided. Proton batteries exhibit specific charge–discharge profiles characterized by sustained voltage plateaus over certain periods. For the discharge characterization, a constant and relatively small current, such as 0.1, 0.5, or 1.0 mA, is applied to the proton batteries to monitor the duration of the voltage plateau. The batteries are charged using the same constant current until they reach their maximum voltage, which is determined by the open-circuit voltage (OCV). The choice of a small charge–discharge current allows researchers to focus on studying the impact of electrolyte and electrode materials on proton battery performance. Table 2 summarizes the research and developments related to PCPE batteries from 2014 to 2023.

### 8.1. Primary Battery

A primary battery, also known as a disposable battery, is designed for single use and cannot be recharged. Its energy output is limited to what can be obtained from the reactants used in its manufacturing. Once the initial supplies of reactants are exhausted, the primary battery cannot be restored or recharged through electrical means. Primary batteries have several advantages, including low cost, high energy density, good shelf life, and minimal maintenance requirements. They find applications in portable electronic devices, photography equipment, toys, and occasionally in high-capacity primary batteries for military and signaling purposes. In the case of proton batteries based on a chitosan–polymer electrolyte with a zinc anode and manganese dioxide (MnO_2_) cathode, their discharge profiles were examined at a constant current of 1.0 mA (Figure 8b). The discharge continued for 17 h until the cut-off voltage of 1.0 V was reached. However, just before reaching a flat discharge plateau at 1.3 V, the voltage of the batteries dropped. This drop was attributed to activation polarization. Activation polarization typically arises from kinetic factors related to charge transfer, such as the activation energy barrier and equilibrium current density. It occurs when the rate of an electrochemical reaction at an electrode surface is limited by slow electrode kinetics [138]. The cathode material, which comes into contact with the electrolyte solution and has a rich surface chemistry, plays a crucial role in activation polarization [139]. Efficient reduction of the cathode is essential for completing the external reaction of proton batteries.

Changing the type of cathode material, specifically from MnO_2_ to V_2_O_5_, in the discharge test at 1.0 mA resulted in different initial voltages: 1.59 V for MnO_2_ and 1.39 V for V_2_O_5_ (Figure 8c). The voltages of both cathode types dropped to 0.50 V after being sustained for 52 min (MnO_2_) and 49 min (V_2_O_5_). However, the discharge capacities of proton batteries containing V_2_O_5_ were lower due to structural conversion caused by mechanical stress during discharge, which led to reduced discharge capacity and operating voltage [140]. The discharge profiles of proton batteries can be influenced by various factors, including the type of salts and plasticizers mixed with the chitosan, as well as the type of polymer electrolyte used. Even though both studies used a similar discharge current (1.0 mA) and had the same type of anode (Zn) and cathode (MnO_2_), the discharge capacities differed between them. These variations highlight the importance of careful selection and optimization of materials for proton batteries to achieve desired performance characteristics.

In another investigation, the discharge current was varied during discharge characterization, and it was observed that the discharge capacity decreased as the discharge current increased [78]. A discharge capacity of 48.0 ± 5.0 mA h was achieved for the proton battery using chitosan/PEO–NH_4_NO_3_–ethylene carbonate as biopolymer electrolytes, which had a conductivity of (2.06 ± 0.39) × 10^−3^ S cm^−1^. However, the low performance of the proton battery in this study may be attributed to one or more of the following reasons:i.electrode/electrolyte: poor contact,ii.the anode condition: inability to supply copiously proton in the PCPE,iii.the cathode condition: undergone structural change during insertion and/or extraction of the proton (developed some interfacial resistance of the cell),iv.conductivity value: lower than ~10^−4^ S cm^−1^.

However, it is important to note that the results obtained by another researcher contradicted this finding [117]. In their study, the discharge capacity increased proportionally with the discharge current. Specifically, a low discharge current of 0.1 mA resulted in a lower specific discharge capacity, even though the discharge time was longer. Conversely, a high discharge current led to a higher specific discharge capacity.

Furthermore, temperature has a significant impact on the characterization of battery discharge capacity. Proton batteries exhibited a greater discharge capacity at 60 °C (1.0 mA) compared to room temperature. At 25 °C, the cell voltage dropped significantly before reaching a flat discharge plateau (1.181 V) due to activation polarization. Although the voltage drop was smaller at a flat discharge plateau (1.368 V, 80 °C), the discharge could only be sustained for 20 h due to the lower discharge capacity (Figure 8d).

### 8.2. Secondary Battery

The role of a PCPE in a secondary battery is to facilitate the migration of H^+^ ions from the anode to the cathode during device discharge and the reverse process during recharging. At elevated temperatures, the movement and diffusivity of H^+^ ions through the cell increased, leading to higher discharge capacity. However, a decrease in discharge capacity was observed at 80 °C due to deterioration of the electrode materials. Some reports conducted charge–discharge tests for those interested in fabricating rechargeable proton batteries (Figure 8e). The charge–discharge characteristics were still dependent on the polymer-electrolyte materials. For example, Zn/chitosan–PVA–NH_4_NO_3_–EC/MnO_2_ proton batteries exhibited charge–discharge cycles lasting nearly 90 hours at a current of 0.3 mA [88]. On the other hand, Zn/starch–chitosan NH_4_Cl-glycerol/MnO_2_ coin-cell proton batteries cycled 40 times and lasted for approximately 440 hours at 0.35 mA [78].

The discharge profile is outlined below:i.Discharge characterization: specific, small constant currents, i.e., 0.1, 0.5, and 1.0 mA, were typically used.ii.Variations in the discharge profile graph and discharge capacity values: modification of several parameters, such as blending a few types of polymers, using GPE, and varying the type of cathode.

## 9. Conclusions and Prospectus/Future/Outlook

The details from previous studies on the characterization of PCPE for solid-state batteries have been reviewed. Here are some key findings:

Proton Movement Mechanism: Understanding the movement of protons within polymer electrolytes is essential for optimizing their performance in solid-state batteries.Ionic Conductivity: The ionic conductivity of PCPEs is influenced by various factors, including their morphology and structure. Enhancing proton conductivity while maintaining stability is crucial.Electrochemical Properties: LSV and CV techniques are used to determine the operational voltage limits of proton–polymer batteries.Thermal Analysis: Thermal stability and degradation characteristics of PCPEs at specific temperature ranges play a critical role in determining the operating temperature range of PCPE batteries.Performance: Although PCPE batteries may have lower performance compared to other electrochemical devices, they are still beneficial for small electronic devices.Polymer Host Modification: Various methods, such as the use of additives, block copolymers, fabrication techniques, and materials, can improve the performance of proton–polymer batteries.

Challenges of PCPE in solid-state batteries include:

Low ionic conductivity: PCPEs often exhibit lower ionic conductivity compared to traditional liquid electrolytes. Enhancing proton mobility within the polymer matrix and optimizing proton transport pathways are ongoing challenges.Chemical stability: PCPEs are prone to chemical degradation, especially under high-temperature and voltage conditions. Developing PCPE materials with improved chemical stability is crucial.Electrode–electrolyte interface compatibility: Ensuring a compatible interface between PCPE electrolytes and electrode materials is critical for efficient charge transfer and overall battery performance.Mechanical stability and flexibility: PCPEs must withstand volume changes and mechanical stresses during charge/discharge cycles while remaining mechanically stable and flexible.Scalability and cost: Developing scalable and cost-effective processes for PCPE synthesis and fabrication is essential for commercial viability.

Future applications of PCPE in solid-state batteries can be broadened by improving several fundamental aspects:

The compilation of all the characterizations into one precise report as a main reference will be beneficial to future researchers as most of the previous studies preferred to focus on a specific type of characterization.Method for preparing the PCPE: varying the physical form of the electrolyte and the materials that are blended with PCPE, i.e., the addition of perovskite proton conductors, can improve the characterization of the electrolyte.Combination of theory and modeling: gain a better knowledge of proton-conducting mechanisms, and the comparison of simulated and real proton–polymer batteries can serve as a benchmark for future research breakthroughs in this field.

## Figures and Tables

**Figure 2 polymers-15-04032-f002:**
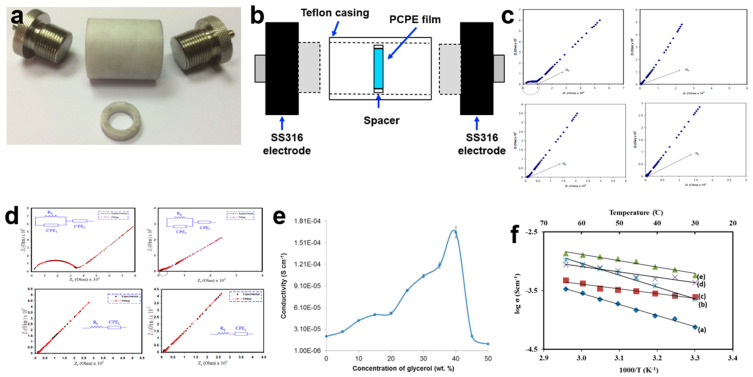
(**a**) Symmetrical blocking electrodes SS316, (**b**) schematic diagram of blocking electrodes inside a Teflon cell, (**c**) Nyquist plot of poly (vinyl alcohol)–ammonium thiocyanate–Cd(II) complex plasticized with glycerol film [81], (**d**) equivalent circuit model of polyvinyl alcohol chitosan ammonium thiocyanate film [82], (**e**) ionic conductivity graph of carboxymethyl cellulose-oleic acid–glycerol film at room temperature [83], and (**f**) temperature dependence graph of carboxymethyl cellulose ammonium acetate [84].

**Figure 3 polymers-15-04032-f003:**
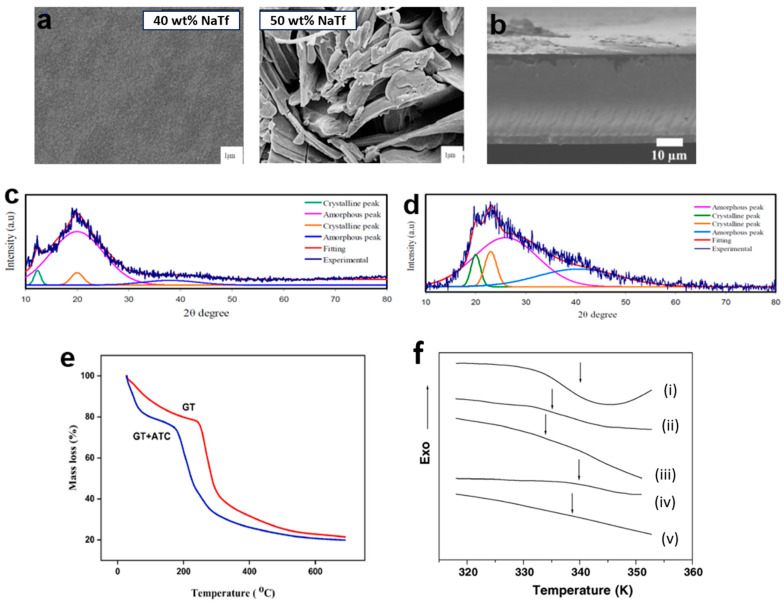
(**a**) SEM surface images of chitosan sodium triflate [93], (**b**) SEM cross-section images for carboxylated chitosan hydrogel–hydrochloric acid [90]. (**c**) X-ray diffraction pattern for chitosan poly(ethylene oxide) 20 wt.% ammonium tetrafluoroborate film [94], (**d**) X-ray diffraction pattern for poly(vinyl alcohol)–ammonium thiocyanate–Cd(II) complex 10 wt.% glycerol [82], (**e**) Thermogram of tragacanth gum-ammonium thiocyanate PCPE [24], (**f**) DSC thermograms for PVA/CH_3_COONH_4_ with concentration NH_4_ +/OH− (i) 0.00, (ii) 0.09, (iii) 0.25, (iv) 0.43 and (v) 0.67 in the vicinity of the glass transition.Adapted with permission from ref [95]. Copyright 2005, Elsevier.

**Figure 6 polymers-15-04032-f006:**
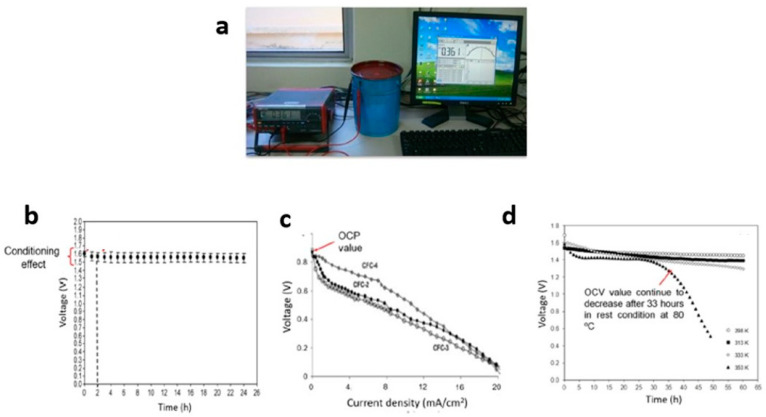
(**a**) True RMS Multimeter (free), (**b**) OCV of (voltage vs. time), (**c**) OCV of (voltage vs. current density), and (**d**) OCV of (voltage vs. time at room and elevated temperature).

**Figure 7 polymers-15-04032-f007:**
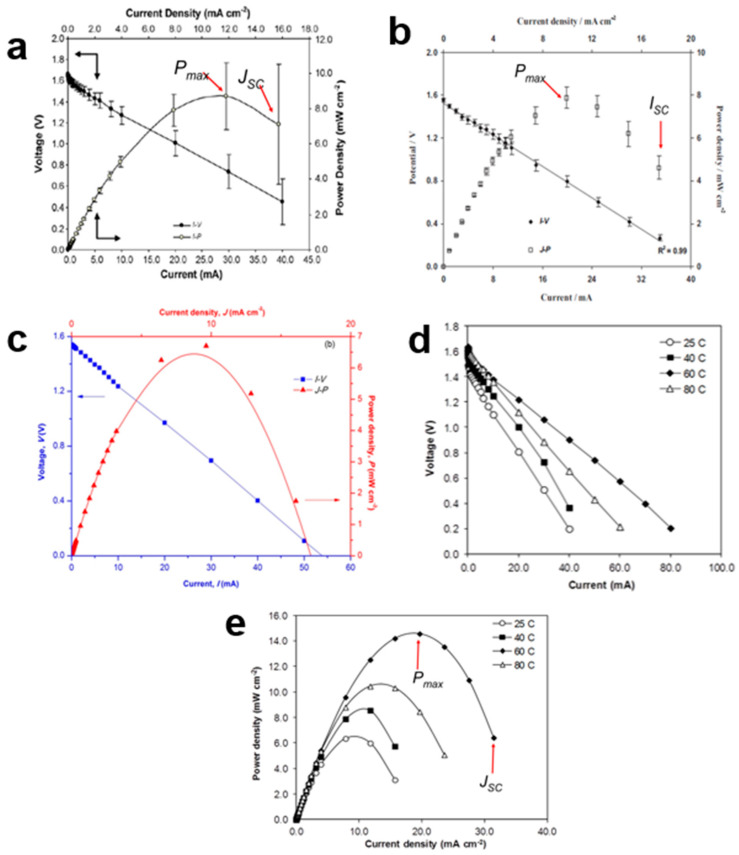
Different plots for proton batteries, (**a**) *I-V* and *J-P* of chitosan NH_4_NO_3_-EC electrolyte (**b**) *I-V* and *J-P* of starch–chitosan NH_4_Cl-glycerol electrolyte (**c**) *I-V* and *J-P* of porous chitosan NH_4_CH_3_COO electrolyte (**d**) *I-V* of chitosan–polymer electrolyte at 25, 40, 60 and 80 °C and. (**e**) *J-P* of chitosan–polymer electrolyte at 25, 40, 60 and 80 °C.

**Figure 8 polymers-15-04032-f008:**
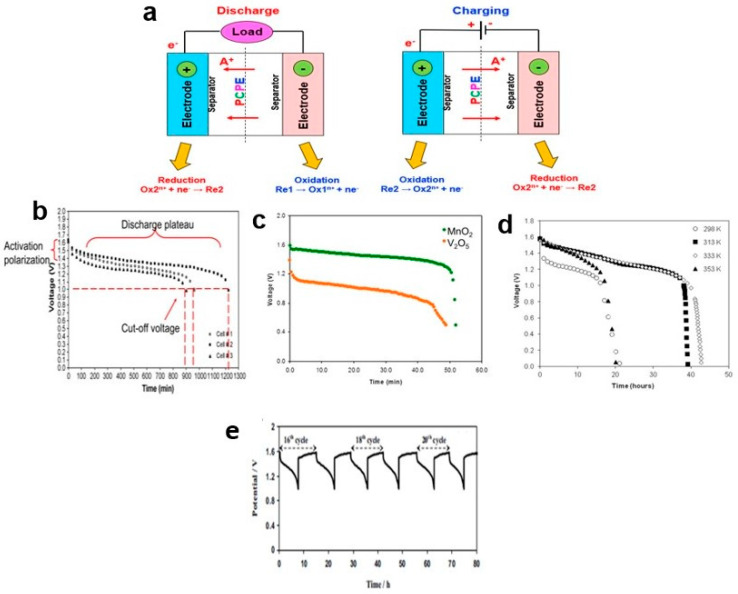
(**a**) Charging and discharging processes in a PCPE battery. (**b**) Discharge curves of proton battery based on chitosan NH_4_NO_3_-EC electrolyte; (**c**) chitosan NH_4_CH_3_COO-EC membranes fabricated with MnO_2_ and V_2_O_5_ cathodes (**d**) chitosan NH_4_NO_3_-ECat 25, 40, 60 and 80 °C (**e**) charge and discharge curves for a proton battery based on Zn/chitosan–PVA–NH_4_NO_3_–EC/MnO_2_ electrolyte and (**e**) charge–discharge profiles of Zn/starch–chitosan NH_4_Cl-glycerol/MnO_2_ coin-cell proton battery at the 16th to 21st cycles, adapted figure from ref [78].

**Table 2 polymers-15-04032-t002:** Summary of selected papers on PCPE batteries from 2014 to 2023.

Characterization	Materials	Scan Rate (mVs^−1^)	Range (V)	Highlights	Ref.
LSV	- Stainless steel|dextran + poly (vinyl alcohol) (PVA) + ammonium nitrate (NH_4_NO_3_)|Stainless steel	1	0–5	- LSV conducted at room temperature - Polymer electrolyte is stable up to 3.32 V (suitable for application in electrochemical devices)	[98]
	- Stainless steel|chitosan: methylcellulose-NH4SCN-42 wt.% glycerol|Stainless steel	10	0–3	- Performed at room temperature - The current increased when the potential reached 2.11 V (the ionic conductivity influenced the decomposition voltage of the polymer electrolyte)	[126]
	- Stainless steel|poly(vinyl alcohol) (PVA): ammonium thiocyanate (NH4SCN): Cd(II) glycerol (Gly)|Stainless steel	10	0–2.5	- Conducted at room temperature - Potential window is seen at 2.1 V (insertion on plasticizer (glycerol) may enhance the electrochemical stability of film)	[82]
CV	Cell 1: SS|poly(vinylidenefluoride-co-hexafluoropropylene) + poly(vinylpyrrolidone) + 1-butyl-3-methylimidazolium hydrogen sulfate, immobilized|SS and Cell 2: Zn + ZnSO_4_·7H_2_O|poly(vinylidenefluoride-co-hexafluoropropylene) + poly(vinylpyrrolidone) + 1-butyl-3-methylimidazolium hydrogen sulfate, immobilized|Zn + ZnSO_4_·7H_2_O	5		- Distinct cathodic and anodic current peaks for Cell 2 whereas no such features are observed for Cell 1 - The anodic and cathodic peaks present in the voltammogram of Cell 2 are separated by ~1.2 V (the use of two electrodes geometry with no reference electrode) - Magnitude of currents: Cell 2 > Cell 1.	[128]
	Cell A: SS|methylcellulose/potato starch ammonium nitrate|SS Cell B: Zn + ZnSO_4_·7H_2_O|methylcellulose/potato starch ammonium nitrate|Zn + ZnSO_4_·7H_2_O			- A combination of ZnSO_4_·7H_2_O and Zn (assumed as an H+ provider) has been employed as an anode in protonic Cell B - The anodic and cathodic peaks divided by ~1.8 V (employment of two-electrode geometry without reference electrode) - Magnitude of currents: Cell B > Cell A (proves the proton conduction)	[129]
OCV—Initial voltage of primary/secondary battery	- Anode: Zn + ZnSO_4_·7H_2_O + C			- The cell exhibited an OCV of 1.62 V	
	- Cathode: PbO_2_ + C + electrolyte + V_2_O_5_				[80]
	- Electrolyte: gellan gum-ammonium thiocyanate				
	- Anode: Zn + ZnSO_4_·7H_2_O - Cathode: PbO_2_ZnSO_4_·7H_2_O + electrolyte + PbO_2_ + V_2_O_5_ - Electrolyte: chitosan acetate 50 wt.% ammonium nitrate			- The OCV characteristic of the cell at room temperature shows an initial voltage of 1.5 V - Dropping to ~1.39 V in the first 11 h of assembly - The cell voltage has been observed to have stabilized and the OCV remained constant at 1.39 V for a period of 8 h	[28]
Discharge Profile—Primary battery	- Anode: Zn + ZnSO_4_·7H_2_O + C - Cathode: PbO_2_ + C + electrolyte + V_2_O_5_ - Electrolyte: iota-carrageenan 0.4 wt.% ammonium nitrate			- The discharge performance (load of 1 MΩ) showed the initial voltage (1.04 V) was dropped to 0.94 V, stabilized for 50 h	[25]
	- Anode: Zn + ZnSO_4_·7H_2_O + C - Cathode: PbO_2_ + C + electrolyte + V_2_O_5_ - Electrolyte: polyvinyl alcohol 0.1 (m.m.%) glycine (0.7 m.m.%) ammonium thiocyanate			- The discharge characteristics at room temperature (load of 1 MΩ) showed the voltage value of the cell remained constant at 1.47 V for 65 h then it decreased to 1.3 V and retained it for 187 h - Beyond the plateau region, the voltage value of the cell drops again	[79]
	- Anode: Zn + ZnSO_4_·7H_2_O + C - Cathode: MnO_2_ + C - Electrolyte: corn starch + polyvinyl pyrrolidone + ammonium bromide			- When the load (100 KΩ) is connected across the cell, the short-circuit current of 46 μA is observed and there is an initial sharp continuous decrement in voltage is obtained for first 3 h - The voltage becomes stabilized at 0.98 V for 112 h (plateau region) - Discharged time: 112 h, Current density: 37.52 μA/cm^2^, Power density: 21.126 mW/Kg, Energy density: 2366.104 mWh/Kg, Discharge capacity: 0.411 μA/h	[137]
	- Anode: Zn + ZnSO_4_·7H_2_O - Cathode: PbO_2_ ZnSO_4_·7H_2_O + electrolyte + PbO_2_ + V_2_O_5_ - Electrolyte: polyvinyl alcohol + amino acid proline + ammonium thiocyanate			- After applying the load (1 MΩ), the stabilized voltage (1.5 V) has dropped to 1.45V which remains constant for 35 h	[99]
Charge–Discharge Profile—Secondary battery	- Anode: Zn + ZnSO_4_·7H_2_O - Cathode: MnO_2_ + electrolyte - Electrolyte: carboxymethyl cellulose 25 wt.% ammonium bromide			- Tested for 10 cycles and showed good rechargeability - The discharge profiles of proton batteries at different constant currents (0.10, 0.25, 0.50 mA) - The longest discharge time with a plateau cell potential close to 1.20 V at the 4th cycle resulting in the highest discharge capacity of 14.61 mA h g^−1^	[31]
	- Anode: Zn + ZnSO_4_·7H_2_O + C + PVdF - Cathode: MnO_2_ + PVdF - Electrolyte: carboxymethyl cellulose ammonium chloride 8 wt.% propylene carbonate			- The discharge characteristics of the cell revealed that the performance was good when it was discharged with 9 µA - The cell showed good rechargeability up to 9 cycles and it was found that the highest discharge capacity was ~2.7 µAh for 20 min	[30]
	- Anode: Zn + ZnSO_4_·7H_2_O - Cathode: PbO_2_ + V_2_O_5_ - Electrolyte: poly (vinylidenefluoride-co-hexafluoropropylene)/poly(vinylpyrrolidone)- 1-butyl-3-methylimidazolium hydrogen sulfate			- The discharge characteristics of the battery at 1 MΩ, 100 kΩ and 10 kΩ loads - 1 MΩ load: the cell remains stable at 1.54 V for ~300 h and provides an energy density of 35.2 W h kg^−1^ - 100 kΩ load: The cell remains stable only up to ~100 h and gives an energy density of 11.3 W h kg^−1^ - 10 kΩ load: The cell remains stable only up to ~ 400 min and gives an energy density of 2.9 W h kg^−1^	[128]
	- Anode: Zn + ZnSO_4_·7H_2_O + C - Cathode: PbO_2_ + C + electrolyte + V_2_O_5_ - Electrolyte: poly (vinylidene fluoride hexafluoropropylene)/poly (methyl methacrylate) + ammonium thiocyanate + ethylene carbonate + propylene carbonate			- The battery was discharged at fixed load resistances of 15 kΩ, 150 kΩ, and 1 MΩ - The performance of the battery is good at low current drain - The discharge capacity cell: 150 kΩ > 1 MΩ > 15 kΩ - The rechargeability of the cell was monitored in the potential range of 1.0–1.4 V through a current of 10 mA for 10 cycles - Quick charge/discharge has been observed for the first three cycles, after which the discharge capacity of the cell fades away significantly (poor intercalation and de-intercalation of H^+^ ions at the cathode) - Possible reasons behind the capacity loss: (i) structural changes in the cathode materials, (ii) the poor interfacial stability of the electrodes	[125]

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
