# Peer review of "A Review of Solid-State Proton–Polymer Batteries: Materials and Characterizations"

_polymers, 2023, doi:10.3390/polym15194032_

Round 1
Reviewer 1 Report
M.S.A. Rani et al. reviewed A Review of Solid-state Proton Polymer Batteries: Materials and Characterizations. The review was mainly focused on the solid-state polymer for proton batteries. The authors have described the challenges of solid-state proton polymer batteries. The topic is highly exciting and the review is very important for the readers.
However, it requires minor revisions in order to meet the journal's requirements.
1. The authors have described several characterization techniques of polymers. However, I feel GPC (Gel permeation chromatography) analysis and its impact on proton conducting polymer was not described.
2. The authors can present the data more systematic way (I feel a table with literature will be interesting). The results can be depicted in a way that readers can correlate how researchers are succeeding to overcome the challenges.
3. How solid-state polymer science is evolving with time in the proton battery is missing.
4. Very recently, researchers are working on di-block, tri-block ionic polymers as a solid-state electrolyte in the proton battery. The authors have not touched on this part. I feel it will be interesting if authors can consider adding block copolymers part in this review.
5. It will be easy for the readers to understand the science if the authors present the chemical structure of the polymers.
Author Response
- Gel permeation chromatography analysis was not our focus of this review. We will plan to include this characterization in another article.
- Table with literature was included in article as suggested by reviewer.
- How solid-state polymer science is evolving with time in the proton battery is added in the article.
- We added block copolymers elaboration as suggested by the reviewer.
- In this review, we will concentrate on recent advances in the characterization technique of choice for proton-conducting polymer electrolytes (PCPE) batteries, including an introduction to proton batteries, electrochemical characterization of materials, and PCPE performance in solid-state batteries. We are not concerned with the polymer itself, but rather with its properties.
Reviewer 2 Report
It was my pleasure to review and evaluate work: “A review of solid-state proton polymer batteries: Materials and Characterizations” (Manuscript ID: polymers-2423299).
Manuscript is well constructed and clearly written. I was unable to find any major editorial or spelling errors. However, there is plenty of minor mistakes which shed negative light on the overall experience of reading proposed text. Presented manuscript has relevant references in introduction and rest of the text. Review is informative. Conclusions are the weakest part of the manuscript I would like to see a valuable discussion of the data.
Presented work is particularly interesting and can be good for the beginners. However I have several remarks and recommendation before publishing this work:
1) Quality of images needs to be improved immensely as some of the graphics are extremely hard to read.
2) Do you acquired permission to copy some of the graphs from other publishers as I think it is necessary step?
3) I would add some interesting findings in similar filed but regarding the supercapacitors (this should be written together not like in the text super capacitors) especially in the introduction for example: Polymers 11 (10), 1648 is interesting work about use of starch in supercapacitors.
4) Line 92 - ZnSO4.7H2O it should be multiplication sign not dot (correct it in whole paper).
5) Line 101 - Scm-1 please work on the unit representation as it can be confusing what unit is presented here is it Sc per meter? Siemens per cm? or maybe something else
6) Line 121 - 7OH- correct the mistakes
7) (Paragraph 3.1) EIS abbreviation was never explained additionally the EIS technique is barley described this needs to be improved to explain to the reader what it is exactly.
8) Line 172 - x-ray diffraction correct spelling
9) In general the characterization techniques of materials of this paper should present more didactic approach somethings are not explained for example: “The deconvolution method is also employed to obtain the XC” and? Electrochemical techniques are well described but why there is no section for EIS or galvanostatic (charge/discharge profile) technique which are more quantitative techniques than voltammetry should be better described and some graphs would be appreciated.
10) Line 244 – “etc” dot is missing
11) It is extremely important to differentiate between voltage and potential. Unfortunately in the text I noticed some confusion especially the graphs.
12) In my opinion figure 8e should be removed as the data presented there is showing more or less a secondary battery which doesn’t work.
13) Part which I dislike the most is the conclusion in my opinion this part needs a complete makeover as it more of summarization than conclusions. Here show deep analyse of what is written in the text.
14) Overall what is lacking in this review is the discussion. At this moment it is more like a report than a review where You comment findings of other authors.
At this point I would recommend presented manuscript to undergo major improvement and can be reconsidered after the mentioned correction.
Manuscript is well constructed and clearly written. I was unable to find any major editorial or spelling errors. However, there is plenty of minor mistakes which shed negative light on the overall experience of reading proposed text.
Author Response
- Quality of images have been improved.
- Permission to copy some of the graphs have been obtained.
- Polymers 11 (10), 1648 work about use of starch in supercapacitors has been added in introduction section.
- Multiplication sign has been replaced throughout the article.
- Line 101 – S cm-1 is Siemens per cm.
- Line 121 - 7OH- has been corrected.
- EIS abbreviation has been explained in Paragraph 3.1.
- Line 172 - x-ray diffraction spelling is corrected.
- Section for EIS or galvanostatic has been added in the manuscript.
- Line 244 – “etc” dot is added.
- Difference between voltage and potential has been revised in the article as well as the figures.
- Figure 8e has been removed as suggested by reviewer.
- The conclusion section has been revised.
- The discussion part has been reviewed and revised.
Round 2
Reviewer 2 Report
After proposed corrections and replies to my comments and suggestions article can be published as it is.